# Anti-spike antibody response to natural SARS-CoV-2 infection in the general population

Jia Wei[1,2], Philippa C. Matthews[1,3], Nicole Stoesser [1,3,4,5], Thomas Maddox[6], Luke Lorenzi[6], Ruth Studley[6], John I. Bell[7], John N. Newton[8], Jeremy Farrar[9], Ian Diamond[6], Emma Rourke[6], Alison Howarth[1,5], Brian D. Marsden[1,10], Sarah Hoosdally[1], E. Yvonne Jones[1], David I. Stuart[1], Derrick W. Crook[1,3,4,5], Tim E. A. Peto[1,3,4,5], Koen B. Pouwels [1,2,4,11,22], A. Sarah Walker[1,2,4,12,22], David W. Eyre [2,3,4,5,22✉] & the COVID-19 Infection Survey team*

Understanding the trajectory, duration, and determinants of antibody responses after SARS-CoV-2 infection can inform subsequent protection and risk of reinfection, however large-scale representative studies are limited. Here we estimated antibody response after SARS-CoV-2 infection in the general population using representative data from 7,256 United Kingdom COVID-19 infection survey participants who had positive swab SARS-CoV-2 PCR tests from 26-April-2020 to 14-June-2021. A latent class model classified 24% of participants as 'non-responders' not developing anti-spike antibodies, who were older, had higher SARS-CoV-2 cycle threshold values during infection (i.e. lower viral burden), and less frequently reported any symptoms. Among those who seroconverted, using Bayesian linear mixed models, the estimated anti-spike IgG peak level was 7.3-fold higher than the level previously associated with 50% protection against reinfection, with higher peak levels in older participants and those of non-white ethnicity. The estimated anti-spike IgG half-life was 184 days, being longer in females and those of white ethnicity. We estimated antibody levels associated with protection against reinfection likely last 1.5-2 years on average, with levels associated with protection from severe infection present for several years. These estimates could inform planning for vaccination booster strategies.

[1] Nuffield Department of Medicine, University of Oxford, Oxford, UK. [2] Big Data Institute, Nuffield Department of Population Health, University of Oxford, Oxford, UK. [3] Department of Infectious Diseases and Microbiology, Oxford University Hospitals NHS Foundation Trust, John Radcliffe Hospital, Oxford, UK. [4] The National Institute for Health Research Health Protection Research Unit in Healthcare Associated Infections and Antimicrobial Resistance at the University of Oxford, Oxford, UK. [5] The National Institute for Health Research Oxford Biomedical Research Centre, University of Oxford, Oxford, UK. [6] Office for National Statistics, Newport, UK. [7] Office of the Regius Professor of Medicine, University of Oxford, Oxford, UK. [8] Health Improvement Directorate, Public Health England, London, UK. [9] Wellcome Trust, London, UK. [10] Nuffield Department of Orthopaedics, Rheumatology and Musculoskeletal Sciences, University of Oxford, Oxford, UK. [11] Health Economics Research Centre, Nuffield Department of Population Health, University of Oxford, Oxford, UK. [12] MRC Clinical Trials Unit at UCL, UCL, London, UK. [22] These authors contributed equally: Koen B. Pouwels, A. Sarah Walker, David W. Eyre. *A list of authors and their affiliations appears at the end of the paper. ✉email: david.eyre@bdi.ox.ac.uk

Till June 2021, over 170 million severe acute respiratory syndrome coronavirus 2 (SARS-CoV-2) infections and over 3 million associated deaths have been reported globally[1]. However, in the months following infection, re-infection is uncommon and anti-spikeSARS-CoV-2 antibodies are associated with protection[2–4]. The duration of post-infection immunity has important implications for the future of the pandemic and vaccination policy[5].

Seroconversion to viral spike and nucleocapsid antigens usually happens within 1–3 weeks after SARS-CoV-2 infection[6–8], with peak antibody levels achieved in 4–5 weeks[9,10]. However, 5–22% of individuals remain seronegative following infection[11–13]. The absence of seroconversion is more common following mild vs. severe disease (e.g., 22.2% vs. 2.6%, $n = 236$[12]) and in asymptomatic vs. symptomatic individuals (11.0% vs. 5.6%, respectively, $n = 2,547$[13]). However, the contribution of other factors, including viral load, has not been comprehensively assessed.

Among those who do seroconvert, data on the trajectory and duration of antibody responses to different SARS-CoV-2 antigens vary, partly reflecting assay-dependent differences even where similar viral antigens are studied[14,15], as well as differences in the populations and disease groups investigated. Estimates for the half-life of anti-spike IgG antibodies (associated with neutralizing activity[16]) vary from 36 to 244 days[15,17–23]. Similarly, anti-nucleocapsid IgG half-lives have been estimated between 35 and 85 days[15,19,20,22].

Most studies have had small to moderate sample sizes or specific sub-populations; large-scale representative population studies are limited. We used the Office for National Statistics (ONS) COVID-19 Infection Survey (CIS), a large community-based survey representative of UK's general population, to investigate predictors of seroconversion following SARS-CoV-2 infection, identify anti-spike IgG antibody trajectories and examine the peak and duration of IgG antibody responses, in particular considering the impact of demographic factors, PCR cycle threshold (Ct) values (inversely related to viral load) and self-reported symptoms on post-infection antibody responses.

## Results

From 26 April 2020 to 14 June 2021, 467,450 participants had one or more throat and nose swab study results (median 10, interquartile range (IQR) 8–12) during a median (IQR) 221 (141–251) days of follow-up. Then, 19,588 (4.2%) participants ≥16 years were ever PCR-positive, 92 (0.5%) with a second episode >120 days after their first PCR-positive result (median 149, IQR 134–174 days later). Analysis included the 7256/19,588 (37%) participants with at least one anti-spike IgG antibody measurement within [−90, +180] days of the start of their first infection episode, who contributed 14,552 antibody measurements (median 2, IQR 1–3, range 1–10; excluding measurements from 3 days after first vaccination) (Supplementary Fig. 1).

The median age of these 7256 participants was 47 (IQR 34–59) years and 3874 (53.4%) were female (Table 1). Next, 6577 (90.6%) reported White ethnicity, 127 (1.8%) working in patient-facing healthcare and 1592 (21.9%) having a long-term health condition. Considering the minimum Ct value across all positive tests in the first infection episode, i.e., the maximum viral load, the median was 27 (IQR 19–32), with 4420 (60.9%) having Ct < 30. Three SARS-CoV-2 PCR target genes (ORF1ab, nucleocapsid protein (N) and spike protein (S)) were tested for in all participants: 1505 (20.7%) were only positive on a single gene (ORF1ab or N) and 2822 (38.9%) were Alpha (B.1.1.7) compatible (i.e., showed S gene target failure). Next, 4190 (57.7%) reported having any symptoms, with 2773 (38.2%) reporting classic symptoms (fever,

cough, loss of smell, or loss of taste). Further, 5169 (71%) participants only contributed antibody measurements after their index positive date.

**Antibody trajectories following SARS-CoV-2 infection.** A latent class analysis identified three classes of post-infection anti-spike IgG antibody responses: Class 1, 'classical seroconversion', Class 2 'possible late detection/re-infection' and Class 3 'seronegative; non-responders'. Class-membership probabilities were high, suggesting that participants' responses could be reliably assigned to one of the three classes (Fig. 1, Supplementary Figs. 2 and 3, and Table 1). Participants who seroconverted after infection comprised Class 1 (N = 4683, 64.5%). These participants showed classical responses, with rises in antibody levels over the 4–5 weeks following their first PCR-positive sample, followed by subsequent waning. Class 1 had lower Ct values (median [IQR] 22 [17–28] vs. Class 2, 32 [30–34], Class 3, 33 [31–34]) and a higher percentage of reported symptoms (77.7% vs. Class 2, 21.8%, Class 3, 21.2%) and classic symptoms (54.8% vs. Class 2, 10.7%, Class 3, 6.7%). Class 1 also had a lower percentage of single gene positives (5.2% vs. Class 2, 34.8%, Class 3, 55.9%). In all, 57.8% had more than one positive swab test in their first infection episode and 23.8% had a positive test in national testing programme prior to their first-study positive test, a significantly higher percentage than other classes (Class 2, 24.4%, 8.1%; Class 3, 3.4%, 1.3%) (Supplementary Table 1).

Class 2 (N = 831 (11.5%), 'possible late detection/re-infection') also had rises in anti-spike IgG levels but these started earlier, before the index positive PCR test. Their antibody levels reached a peak around the time of the index positive and then waned. This class likely partly reflects the study design, as study PCR testing was conducted at regular, usually monthly, intervals, irrespective of symptoms, with a proportion of missed visits (see 'Methods'). Therefore, this group could represent those where infection was detected late rather than reflecting any underlying biological difference. However, a subset may also represent re-infection with an undetected first infection. Supporting these possibilities, Ct values were higher (median [IQR] 32 [30–34]) than Class 1 and self-reported symptoms were less common (21.8%), as were multiple positive PCR tests (24.4%) (Table 1 and Supplementary Table 1). For more participants, the index positive PCR was their first test in the study (27.4%); in the remainder, the median days since last negative was 29 days, higher than other classes and with considerable skew, with 369 (44.4%) being >31 days and 256 (30.8%) being >59 days, supporting late detection contributing to this group.

Lastly, 1742 (24.0%) participants were assigned to Class 3 ('seronegative; non-responders'). Their IgG levels barely increased and were below the positivity threshold throughout (excepting 17 outlier individuals who appeared to mount a response >30 days after their index positive PCR test). Compared with Class 1, Class 3 had higher Ct values (median [IQR] 33 [31–34]), a lower percentage self-reporting symptoms (21.2%) or classic symptoms (6.7%) (Table 1). Very few had more than one positive swab in their first infection episode (3.4%) or an accompanying positive test in the national testing programme (1.9%) (Supplementary Table 1). Although this class would be expected to be enriched for false positives, of 1742 participants in this class, 595 (34%) still had strong evidence for a true-positive PCR result (Ct ≤ 32 and ≥2 genes detected). Of the 5230 participants with Ct ≤ 32 and ≥2 genes detected, 595 (11%) were non-responders, 373 (7%) in Class 2 and 4259 (81%) in Class 1. Class 3 were also older (Supplementary Fig. 2), with fewer patient-facing healthcare workers (0.8%) and more participants with long-term health conditions (25.0%).

**Table 1 Main characteristics of participants in classes identified from latent class mixed models for 7256 participants infected with SARS-CoV-2.**

| | Total (N = 7256) | Class 1 'seroconverted' (N = 4683) | Class 2 'possible late/re-infection' (N = 831) | Class 3 'seronegative non-responders' (N = 1742) | p-Value (all) | P-value (2 vs. 1) | P-value (3 vs. 1) |
|---|---|---|---|---|---|---|---|
| Percentage | 100% | 64.5% | 11.5% | 24.0% | | | |
| Age | | | | | <0.001 | 0.006 | <0.001 |
| Median | 47 | 46 | 44 | 51 | | | |
| IQR | 34, 59 | 33, 58 | 32, 56 | 36, 65 | | | |
| Sex | | | | | 0.6 | 0.9 | 0.3 |
| Female | 3874 (53.4%) | 2485 (53.1%) | 440 (52.9%) | 949 (54.5%) | | | |
| Male | 3382 (46.6%) | 2198 (46.9%) | 391 (47.1%) | 793 (45.5%) | | | |
| Ethnicity | | | | | 0.001 | 0.2 | 0.002 |
| White | 6577 (90.6%) | 4224 (90.2%) | 738 (88.8%) | 1615 (92.7%) | | | |
| Non-white | 679 (9.4%) | 459 (9.8%) | 93 (11.2%) | 127 (7.3%) | | | |
| Report working in patient-facing healthcare | | | | | 0.002 | 0.4 | <0.001 |
| No | 7129 (98.2%) | 4590 (98.0%) | 811 (97.6%) | 1728 (99.2%) | | | |
| Yes | 127 (1.8%) | 93 (2.0%) | 20 (2.4%) | 14 (0.8%) | | | |
| Report having long-term health condition | | | | | <0.001 | 0.002 | 0.004 |
| No | 5664 (78.1%) | 3668 (78.3%) | 690 (83.0%) | 1306 (75.0%) | | | |
| Yes | 1592 (21.9%) | 1015 (21.7%) | 141 (17.0%) | 436 (25.0%) | | | |
| Minimum Ct value across the infection episode | | | | | <0.001 | <0.001 | <0.001 |
| Median | 27 | 22 | 32 | 33 | | | |
| IQR | 19, 32 | 17, 28 | 30, 34 | 31, 34 | | | |
| Ct positivity pattern (S-gene positivity) | | | | | <0.001 | <0.001 | <0.001 |
| ORF + S or N + S or ORF + N + S | 2929 (40.4%) | 2361 (50.4%) | 221 (26.6%) | 347 (19.9%) | | | |
| ORF + N | 2822 (38.9%) | 2079 (44.4%) | 321 (38.6%) | 422 (24.2%) | | | |
| ORF or N only | 1505 (20.7%) | 243 (5.2%) | 289 (34.8%) | 973 (55.9%) | | | |
| Self-reported symptoms | | | | | <0.001 | <0.001 | <0.001 |
| No | 3066 (42.3%) | 1044 (22.3%) | 650 (78.2%) | 1372 (78.8%) | | | |
| Yes | 4190 (57.7%) | 3639 (77.7%) | 181 (21.8%) | 370 (21.2%) | | | |
| Self-reported classic symptoms | | | | | <0.001 | <0.001 | <0.001 |
| No | 4483 (61.8%) | 2115 (45.2%) | 742 (89.3%) | 1626 (93.3%) | | | |
| Yes | 2773 (38.2%) | 2568 (54.8%) | 89 (10.7%) | 116 (6.7%) | | | |
| Posterior class-membership probability (%) | | | | | <0.001 | <0.001 | <0.001 |
| Median | 95 | 93 | 72 | 100 | | | |
| IQR | 80, 100 | 80, 98 | 60, 99 | 100, 100 | | | |

Additional characteristic comparisons among classes are presented in Supplementary Table 1 and continuous variables are presented graphically in Supplementary Fig. 2. Continuous variables were compared using Kruskal–Wallis tests and categorical variables were compared using one-sided $\chi^2$-tests. ORF: ORF1ab gene.

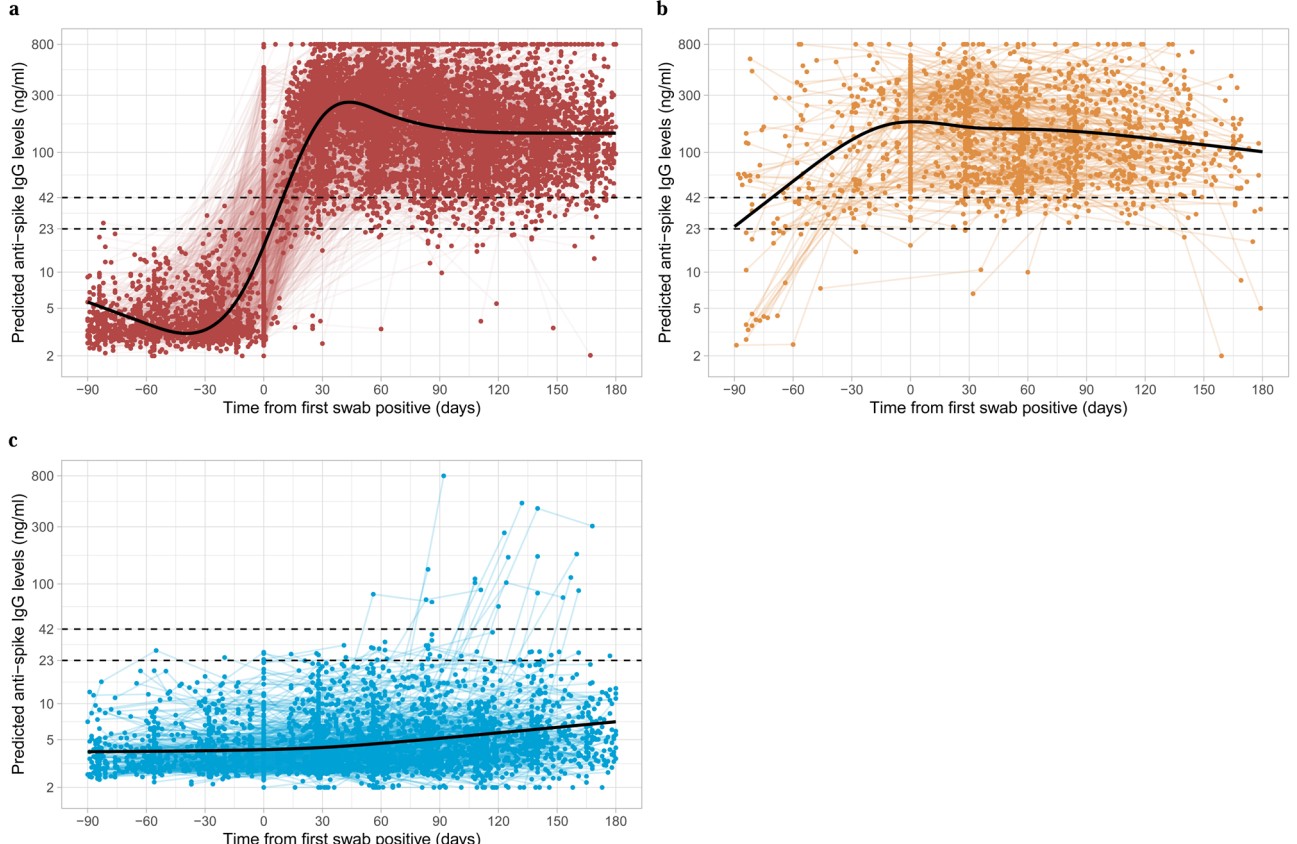

**Fig. 1 Individual trajectories for 7256 participants infected with SARS-CoV-2 by class identified from latent class mixed models. a** Class 1, 'seroconverted in response to infection' ($N = 4683$, 64.5%). **b** Class 2, 'possibly late/re-infection' ($N = 831$, 11.5%). **c** Class 3, 'seronegative non-responders' ($N = 1742$, 24.0%). Black dashed line indicates the assay threshold for IgG positivity (42 ng ml$^{-1}$) and the dotted line at 28 ng ml$^{-1}$ (indicates level associated with 50% protection against re-infection). Restricted natural cubic splines (internal knots at −10, 30, 60 days and boundary knots at −60 and 140 days) were used to model time (see 'Methods'). Distribution of the factors by class membership is shown in Table 1.

**Predictors of non-response**. In the multinomial logistic regression model, independent predictors of remaining seronegative (Class 3) vs. seroconverting (Class 1) were higher minimum Ct (i.e., lower viral load), not self-reporting symptoms, older age and not working in patient-facing healthcare (Fig. 2 and Supplementary Table 2), with no evidence of independent effects of sex, ethnicity or long-term health conditions. For example, at the median age of 47 years (not working in patient-facing healthcare), the Ct threshold at which seroconversion rates reached >90% were 26, 23 and 17 for those reporting classic symptoms, other symptoms or no symptoms, respectively (Fig. 2b). Excluding Ct from the model, there was still no evidence of independent effects of long-term health conditions, but non-White ethnicity was associated with lower odds of being in Class 3 (odds ratio (OR) = 0.70, 95% confidence interval (95% CI) 0.55–0.90, $p = 0.005$) than Class 1.

To investigate associations with specific symptoms, we fitted a logistic regression model comparing only seroconversion (Class 1) vs. non-response (Class 3) and omitting Ct and other test characteristics, as these may mediate effects of symptoms. We found cough, loss of smell, fever, loss of taste, fatigue, headache and sore throat were associated with lower odds of non-response, with cough (OR = 0.20, 95% CI 0.15–0.25, $p < 0.001$) and loss of smell (OR = 0.21, 95% CI 0.13–0.33, $p < 0.001$) most strongly associated. Results remained similar, restricting seronegatives to those with stronger evidence of a true PCR-positive result (Ct ≤ 32 and ≥2 genes detected) (Fig. 3 and Supplementary Table 3, with non-linear effect of age in Supplementary Fig. 4). We additionally

examined the association with specific comorbidities by incorporating them into the model but found no strong evidence of major impact (Supplementary Table 4).

**Determinants of the peak and half-life of antibody responses**. In those who showed a classical antibody response, i.e., Class 1, we estimated anti-spike IgG peak antibody levels and half-life post infection. Those in Class 2 where the timing of first infection was unclear and those who remained seronegative in Class 3 were not included, because their antibody trajectories followed different patterns (Fig. 1). We estimated trajectories from 56 days after the first positive in the infection episode, when the IgG levels were close to the maximum level with high data completeness (Supplementary Fig. 5). Then, 3271 participants were included in this analysis, contributing 5148 antibody measurements (interval censored at an assay upper limit of 800 ng ml$^{-1}$ mAb45 equivalent units), median (IQR) [range] 1 (1–2) [1–5] per participant. Using a Bayesian linear mixed model, assuming antibody levels fell exponentially (i.e., linearly on the log scale, in line with previous studies[17,24,25]) and accounting for variation in individuals' peak levels and half-lives using correlated random effects, the estimated mean anti-spike IgG half-life was 184 days (95% credibility interval (95% CrI) 163–210) and peak level was 203 ng ml$^{-1}$ (95% CrI 190–210) (Fig. 4). Estimated peak levels varied substantially between participants, ranging from 42 to 1390 ng ml$^{-1}$ (Supplementary Fig. 6a). Longer half-lives were correlated with lower peak levels (Supplementary Fig. 6b)

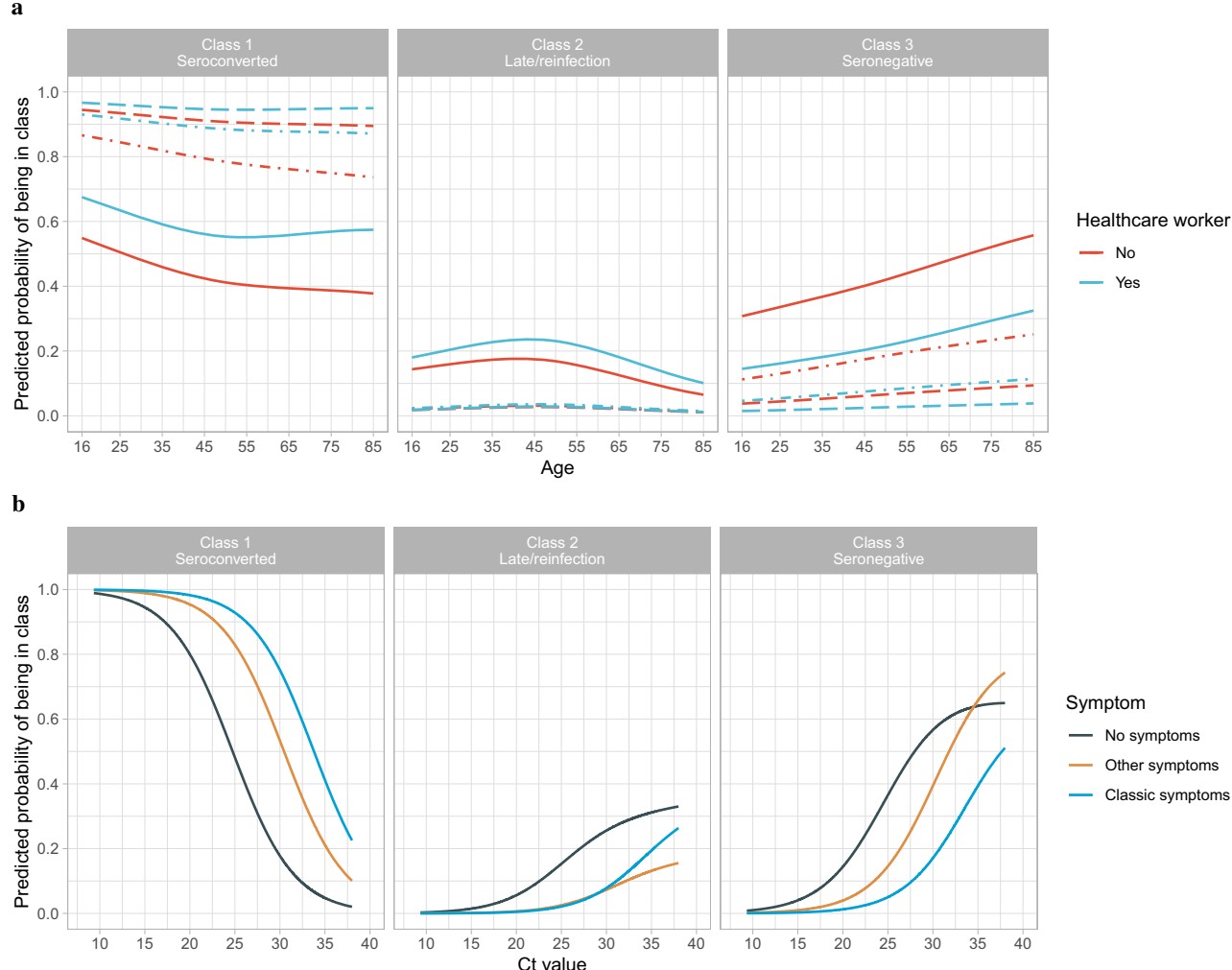

**Fig. 2 Predicted probability of being in Class 1 (seroconverted in response to infection), 2 (possible late/re-infection) and 3 (seronegative non-responders). a** By age and working in patient-facing healthcare, plotted at the reference category for other variables (female, White ethnicity, no long-term health condition, Ct = 26, have only one positive swab test during the infection episode) and no symptoms (solid line), other symptoms (dash-dotted line), classic symptoms (dashed line). **b** By Ct value and self-reported symptoms, plotted at the reference category for other variables (47-year-old, female, White ethnicity, no long-term health condition, not working in patient-facing healthcare, have only one positive swab test during the episode). Age was fitted using natural cubic spline with one internal knot placed at 50 years and two boundary knots at 20 and 80 years. Full model results are shown in Supplementary Table 2.

(Spearman's rank coefficient = −0.50, $p < 0.0001$; correlation between random intercept and slope −0.26). Results were similar in sensitivity analyses starting modelling from different times and using different interval censoring thresholds (400 and 500 ng ml$^{-1}$) (Supplementary Table 5).

In the multivariable linear mixed model, age, ethnicity and Ct values were independently associated with IgG peak levels (model intercept), whereas sex and ethnicity were independently associated with IgG half-life (model slope) (Table 2, Supplementary Table 6 and Supplementary Fig. 7; posterior checks and Markov chain Monte Carlo (MCMC) diagnostics in Supplementary Table 6 and Supplementary Figs. 8 and 9). Conditional on having seroconverted (which occurred at lower rates in older individuals), older age was associated with higher IgG peak levels (adjusted 18 ng ml$^{-1}$ higher (95% CrI 13–23) per 10 years older). Males had a shorter half-life than females (adjusted 77 days shorter, 95% CrI 23–178). Non-White participants had higher IgG peak levels (adjusted 82 ng ml$^{-1}$ higher, 95% CrI 55–113) than White participants, but a shorter half-life (adjusted 75 days shorter, 95% CrI 1–181). Higher Ct values (i.e., lower viral burden) were associated with a slightly higher peak

level (adjusted 1 ng ml$^{-1}$ higher (95% CrI 0–2) per 1 unit higher). Conditional on inclusion in the analysis, i.e., seroconversion, we did not find any evidence of effect of reported long-term health conditions or self-reported symptoms on either IgG peak levels or half-life. In a sensitivity analysis, we did not find effects of time period (pre-Alpha vs. Alpha) on either IgG peak levels or half-life (Supplementary Table 7).

**Duration of antibody responses and possible associated immune protection.** Multivariable linear mixed models allowing for a biphasic exponential decline, or flexible non-linear decline, in antibody levels showed evidence of better model fit than the baseline model using exponential decline (Supplementary Fig. 10). Over the 120 days following peak antibody levels, all three models were qualitatively similar (Fig. 4); however, the more flexible models showed the rate of decline in antibodies slowing over time. We therefore used a multivariable biphasic exponential mixed model to estimate that antibody responses were likely to remain positive, i.e., ≥42 ng ml$^{-1}$, for 703 (95% CrI

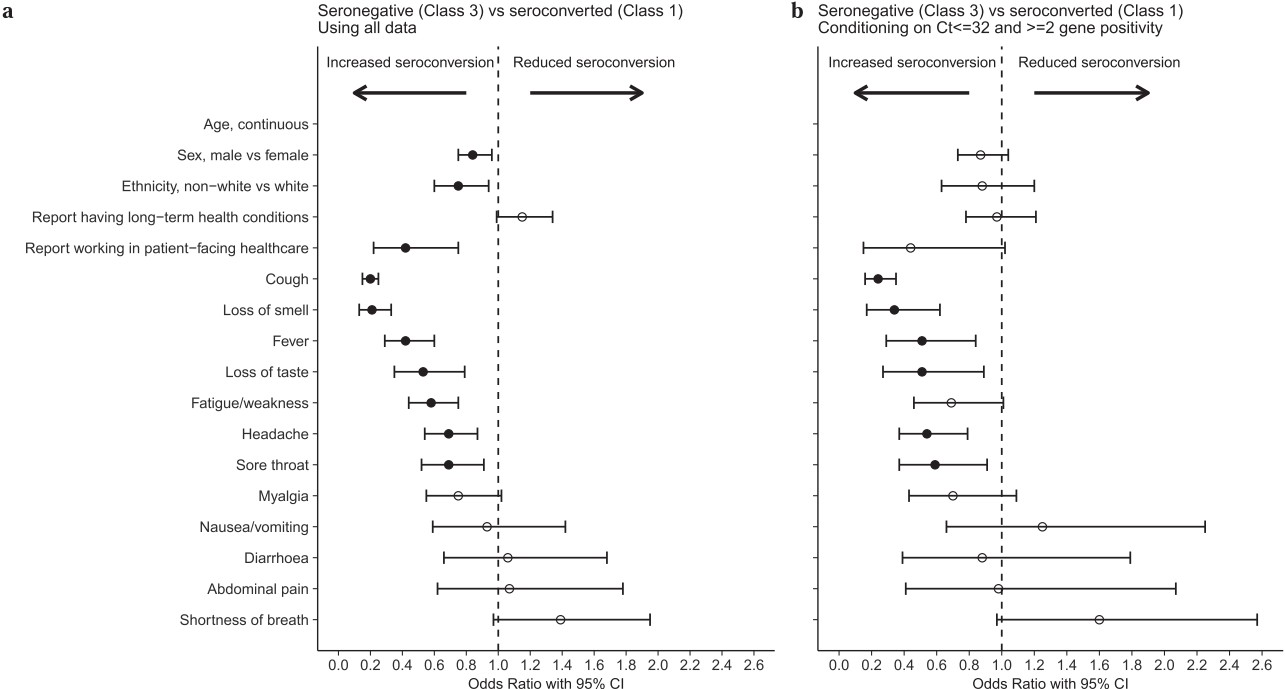

**Fig. 3 Odds ratio with 95% confidence intervals from logistic regression comparing seronegative vs. seroconverting (Class 3 vs. Class 1) using demographic factors and individual symptoms that would be available without a positive test result. a** Using all data from Class 3 (N = 1742) vs. Class 1 (N = 4683). **b** Restricting Class 3 to those with Ct value ≤ 32 and ≥2 genes detected (N = 595) to decrease the impact of potential false-positive swab tests. Age was fitted using natural cubic spline with one internal knot placed at 50 years and two boundary knots at 20 and 80 years. Effect of age is presented in Supplementary Fig. 4. The 95% confidence intervals are calculated by prediction ± 1.96 × SE of the prediction; solid dots indicate estimates with p-values < 0.05, whereas hollow dots indicate those with p-values ≥ 0.05. Numbers of odds ratio, 95% CI and p-values are presented in Supplementary Table 3.

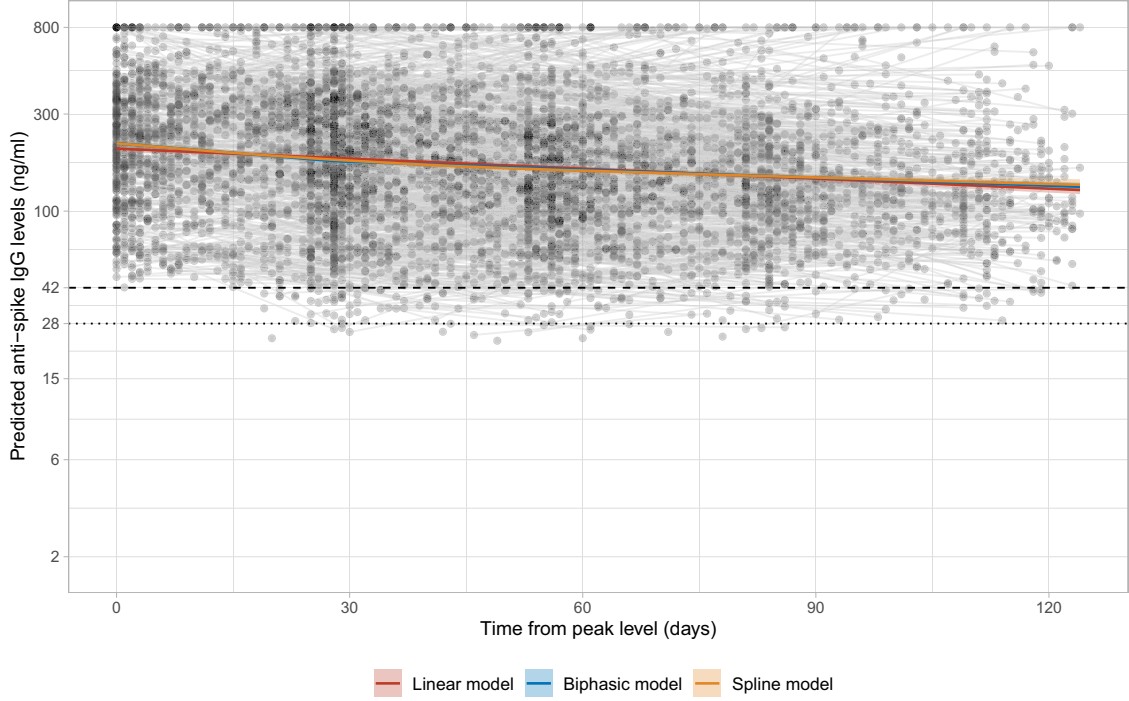

**Fig. 4 Estimated mean trajectory of anti-spike IgG antibody levels and individual trajectories in 3271 participants in Class 1.** The timing of the peak level 56 days after the first positive swab is determined from the latent class mixed model. Estimated trajectories from three models are presented: the model assuming a linear decline in $\log_2$ scale (red line), the biphasic exponential model (blue line), and the model using splines (orange line). For the biphasic model, knot is placed at 28 days. For the spline model, time is fitted using natural cubic splines with internal knots at 30, 70 and boundary knots at 5, 110. The posterior mean and 95% credibility interval are shown. Black dashed line indicates the assay threshold for IgG positivity (42 ng ml$^{-1}$) and the dotted line indicates level associated with 50% protection against re-infection (28 ng ml$^{-1}$).

**Table 2 Posterior mean and 95% credibility intervals for anti-spike IgG peak level (intercept) (ng ml$^{-1}$) and half-life (slope) (days) in the univariable and multivariable models in 3271 participants in Class 1.**

| | | Univariable model | | | Multivariable model | | |
|---|---|---|---|---|---|---|---|
| | | Posterior mean | 95% CrI | | Posterior mean | 95% CrI | |
| Baseline | Peak level (Intercept) (ng ml$^{-1}$) | 203 | 190 | 210 | 185 | 157 | 201 |
| | IgG half-life (slope) (days) | 184 | 163 | 210 | 233 | 161 | 364 |
| Age | Peak level: 43 years (median) | 200 | 187 | 206 | | | |
| | IgG half-life: 43 years (median) | 204 | 177 | 237 | | | |
| | Change in peak level: per 10-year older | 17 | 13 | 22 | 18$^a$ | 13 | 23 |
| | Change in half-life: per 10-year older | −2 | −20 | 21 | −8 | −36 | 23 |
| Sex | Peak level: female | 199 | 183 | 209 | | | |
| | IgG half-life: female | 232 | 189 | 291 | | | |
| | Change in peak level: male | 8 | −6 | 21 | 6 | −7 | 19 |
| | Change in half-life: male | −79 | −141 | −30 | −77$^a$ | −178 | −23 |
| Ethnicity | Peak level: White | 197 | 184 | 204 | | | |
| | IgG half-life: White | 190 | 166 | 219 | | | |
| | Change in peak level: non-White | 70 | 40 | 99 | 82$^a$ | 55 | 113 |
| | Change in half-life: non-White | −46 | −92 | 13 | −75$^a$ | −181 | −1 |
| Long-term health conditions | Peak level: no | 198 | 182 | 206 | | | |
| | IgG half-life: no | 186 | 163 | 213 | | | |
| | Change in peak level: yes | 24 | 6 | 43 | 13 | −2 | 30 |
| | Change in half-life: yes | 11 | −50 | 100 | 10 | −93 | 162 |
| Ct value | Peak level: 22 (median) | 202 | 190 | 209 | | | |
| | IgG half-life: 22 (median) | 184 | 163 | 209 | | | |
| | Change in peak level: per 1 unit higher | 1 | 0 | 2 | 1$^a$ | 0 | 2 |
| | Change in half-life: per 1 unit higher | 0 | −4 | 4 | 0 | −7 | 7 |
| Symptom | Peak level: no | 205 | 180 | 221 | | | |
| | IgG half-life: no | 154 | 124 | 197 | | | |
| | Change in peak level: yes | −3 | −20 | 14 | 1 | −14 | 16 |
| | Change in half-life: yes | 41 | −8 | 86 | 57 | −64 | 172 |

The reference category in the multivariable model is: 43-year-old, female, White ethnicity, no long-term health conditions, Ct value = 22, and no self-reported symptoms.
$^a$Where the multivariable 95% CrI excludes 0 (no effect).

371–2654), 490 (286–923), 561 (334–1456) and 441 (285–809) days from the start of infection for White females, White males, non-White females and non-White males aged 60 years, respectively. From the start of infection to 28 ng ml$^{-1}$, the antibody level associated with 50% protection against new infection in a study of those previously infected[4], the estimated time was 869 (482–3145), 600 (376–1123), 667 (407–1710) and 520 (343–962) days, respectively. For a threshold of 6 ng ml$^{-1}$, estimated to provide 50% protection against severe infection (based on previous estimates that this was provided by neutralizing antibody levels 3% of peak[25]), the estimated time was 1500 (871–5973), 1017 (685–1909), 1070 (669–2827) and 826 (571–1532) days, respectively (Fig. 5 and Supplementary Table 8). To allow for emerging viral variants needing higher antibody concentrations to afford the same level of neutralizing activity, a sensitivity analysis assumed two- to tenfold greater antibody concentrations were required. For example, if fivefold higher concentrations were required, for an example 60-year-old White male, the estimated duration of response was 71 (57–247), 162 (57–391) and 581 (361–1093) days for levels associated with a positive result (5 × 42 ng ml$^{-1}$), 50% protection from infection (5 × 28 ng ml$^{-1}$), and 50% protection against severe infection (5 × 6 ng ml$^{-1}$), respectively. We also presented the estimates of duration of protection using the linear exponential model (Supplementary Fig. 11 and Supplementary Table 8), which yielded shorter estimates of durations for each population group.

## Discussion

We use data from a representative national UK survey to determine predictors of seroconversion following a positive PCR test

and investigate the duration of antibody responses and possible associated protection in those who do seroconvert.

We found 24% of participants did not seroconvert after testing PCR positive. However, if we restricted to participants with strong evidence for a true-positive PCR result (Ct ≤ 32, ≥2 genes detected), a lower proportion, 595/5230 (11%), did not seroconvert. Similar observations have been reported before, but with varying percentages of non-responders from 0% to 25%[11–13,26–29]. Non-responders likely reflect a combination of genuine non-responders, false-positive PCR results, and false-negative antibody results. However, those not seroconverting typically had persistently low antibody levels, suggesting that any false-negative antibody results would not change if antibody positivity thresholds were adjusted within reasonable limits. Consistent with the first two possibilities, non-responders had fewer symptoms and higher Ct values (lower viral loads) but, more consistent with being genuine non-responders, they were also older. The sensitivity of the serological assay was previously reported as 99%[30], based on samples from predominantly symptomatic individuals, including those admitted to a hospital, and it is possible that the greater proportion of those with asymptomatic or mild infection explains part of the difference seen here. We found no evidence of an independent effect of long-term health conditions on non-response, possibly reflecting the heterogeneity of this group including those with a range of cardiovascular and metabolic conditions not typically associated with impaired humoral immunity, as well as conditions more directly impacting antibody production. Other studies have reported that people taking immunosuppressive medications or with impaired immunity have decreased antibody responses[31–35]. Although in some populations antibodies are associated with

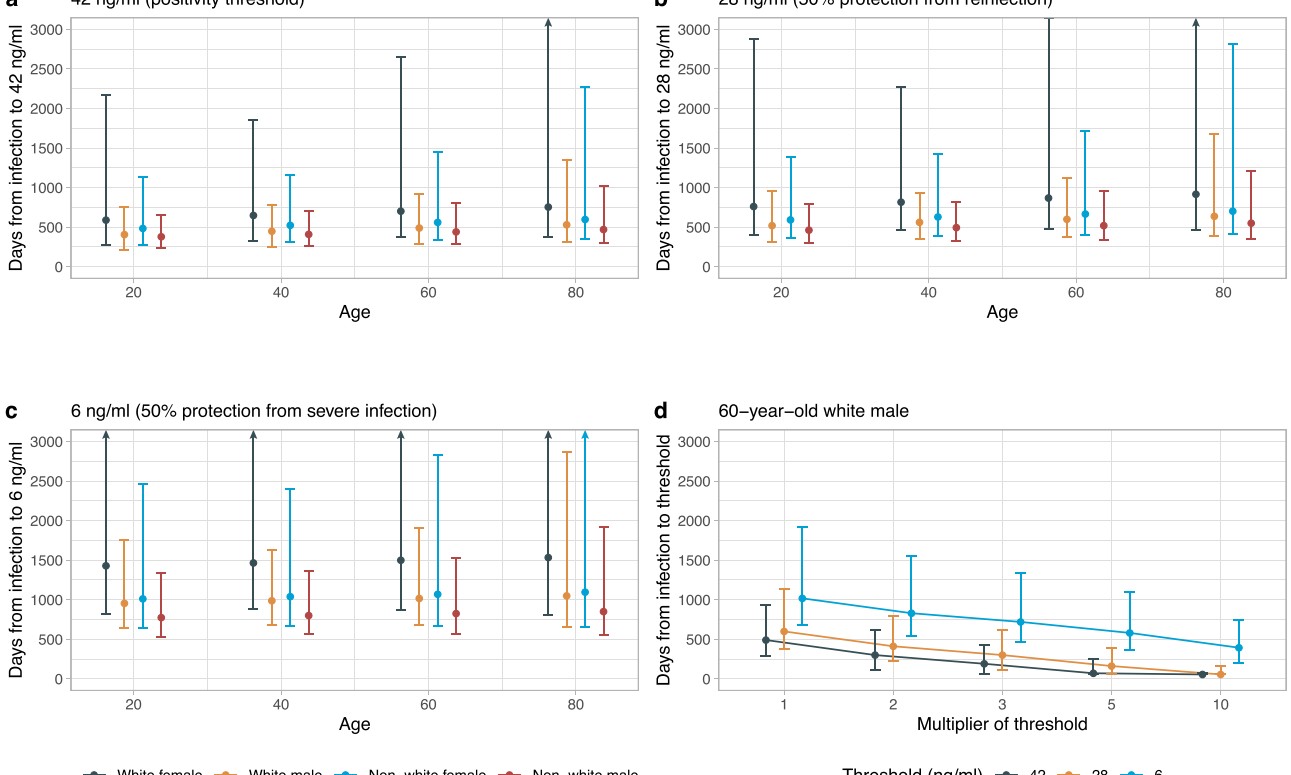

**Fig. 5 Posterior predicted time (95% credibility interval) of the start of infection to three anti-spike IgG thresholds (42, 28 and 6 ng ml⁻¹) by age (20, 40, 60 and 80 years), sex, and ethnicity from the multivariable biphasic exponential model in 3271 participants. a** Time from the start of infection to the positivity threshold of 42 ng ml⁻¹. **b** Time from the start of infection to the equivocal threshold of 28 ng ml⁻¹, which corresponds to 50% protection against PCR-confirmed re-infection. **c** Time from the start of infection to 6 ng ml⁻¹, which corresponds to 50% protection against severe infection. **d** Time from the start of infection to the above three thresholds multiplied by 2, 3, 5 and 10, in a 60-year-old White male as an example, to estimate the duration given the higher antibody level required for protection against variants of concern. Estimations are shown in Supplementary Table 8. Estimates using the linear exponential model are shown in Supplementary Fig. 11 and Supplementary Table 8.

protection from re-infection[3,4], the risk of re-infection and vaccine failure in PCR-positive seronegative individuals from specific immunocompromised groups needs further study.

Although the specificity of PCR testing in this cohort has been estimated as ≥99.995%[36,37], given the large number of tests performed in asymptomatic individuals, i.e., with a low pre-test probability of infection, assuming a sensitivity of 94%[38] and specificity 99.995%, the positive predictive value of PCR tests ranges between 95.0% and 99.7% for SARS-CoV-2 prevalences between 0.1% and 2%. Therefore, although some non-responders will have had false-positive PCR results, in particular as the majority (97%) of participants in Class 3 have only one positive swab test during the study, the relatively high PPV suggests most non-responders are likely to have had a true-positive PCR result. It is also possible that some previous infections were missed by PCR testing, e.g., because participants were not tested, variation in self-performed swabbing technique, or the assay itself.

We found that apart from age, individual symptoms including cough, loss of smell/taste, fever, fatigue, headache and sore throat were independently associated with generating antibodies following a positive PCR test. The strongest predictors were the four classic symptoms (cough, loss of smell/taste and fever).

We estimated the half-life of anti-spike IgG to be 184 days, indicating a sustained antibody response against infection, compared with previous reports between 36 and 244 days[15,17,19–23]. We found multiple factors associated with peak levels and decline. Variation in the literature may be explained by differences in study design, population (age and sex) and assay

performance (different targets and assay types). Longer half-lives were correlated with lower peak levels, suggesting some individuals, e.g., after mild disease[20,28], mount a lower antibody response that wanes more slowly, whereas others produce higher antibody responses but that wane more quickly. This contrasts with a previous healthcare worker study that found a positive correlation between IgG half-life and peak levels[15], but agrees with another reporting a faster decay of IgG in hospitalized patients with high initial responses than individuals with asymptomatic or mild infections[23]. As most SARS-CoV-2 infection is mild/asymptomatic, the duration of antibody responses in our study are likely to best generalize to the population at large.

As expected from previous studies of humoral immunity, older age was associated with lower seroconversion rates. However, among those that did seroconvert, peak IgG levels were higher in older individuals. Similar findings have been reported in healthcare workers, where older age (in those of working age) was associated with higher maximum anti-nucleocapsid IgG levels and longer half-lives[15]. Others have also reported associations between older age and higher immune responses, including IgG and memory B cells[39]. In our study, selection bias may contribute, as our findings are conditional on participants seroconverting and the subset of older participants who seroconvert may have more robust immune responses than younger participants overall, among whom more may seroconvert despite more heterogenous underlying immunity.

Females previously infected with SARS-CoV-2 have been found to have more robust T-cell activation and develop stronger

antibody responses than males[40,41]. We found that males were equally likely to seroconvert; however, among those who did seroconvert, males had a shorter IgG half-life than females, despite no evidence of difference in peak IgG levels, consistent with a previous healthcare worker study[42]. Another study found no difference in IgG antibody between males and females in mild infection and recovering patients, but a higher IgG in females than males in severe infections and early phases of infection[43]. We found non-White participants were more likely to seroconvert than White participants (in models not adjusting for Ct value) and to develop higher antibody levels that then waned more quickly. Higher antibody levels in individuals of non-White ethnicity have been reported in several healthcare worker populations[15,44]. The observed sex and ethnicity effects likely arise from a combination of genetic and societal factors, with further adjustment for confounding arising from social differences and structural inequalities required to estimate the relative contributions of each mechanism.

Although lower Ct values were associated with seroconversion, we found that higher Ct values were associated with slightly higher peak IgG levels, which was counterintuitive, as higher Ct values (lower viral burden) have been previously associated with lower antibody titres[8,20,45]. The most likely explanation is that as testing was conducted at regular intervals, rather than in response to symptoms, measured Ct values do not fully reflect peak viral load in our study. We found no evidence of association between self-reported symptoms and IgG peak levels or half-life, although symptoms were associated with seroconversion; previous findings suggest that symptomatic infections develop stronger antibody responses than asymptomatic infections[27]. This could be because our models conditioned on those who seroconverted or because infections in this general population were generally mild.

Important findings from our study are the predictions about the duration of antibody responses associated with protection from infection, albeit that these related to thresholds previously associated with protection from re-infection or protection from severe infection in vaccine trials. Other immune responses may last for differing time periods and also memory responses may mean that protection lasts longer than measurable antibody levels. Furthermore, consistent with previous studies suggesting antibody waning following a biphasic exponential pattern[46,47], we observed that the rate of antibody decline slows over time and antibody levels can be sustained for longer than assuming antibody levels fall exponentially. We estimated the time from peak level to three thresholds, the positivity threshold 42, 28 (50% protection from any symptomatic/asymptomatic infection[4]) and 6 ng ml$^{-1}$ (3% of our estimated peak level, providing 50% protection against severe infection according to ref. [25]). Based on extrapolations from other studies correlating anti-spike IgG antibody titres with neutralizing activity and early protection (i.e., within a year) from re-infection with currently circulating variants, we found that 50% protection against infection might be expected to last 1.5–2 years, with protection against severe infection potentially lasting several years, but with uncertainty in the precise estimates given the assumptions relating neutralizing activity to antibody titres over time and in estimating levels associated with protection against severe infection[25]. However, given that variants may require higher antibody levels for the same level of neutralization, the duration of protection might be substantially reduced. It may also be the case that the functional quality of antibodies changes over time[48]; this was not evaluated in this study. Overall, at least in the short-term, protection against re-infection appears high.

Study limitations include the fact that we only measured anti-spike IgG using a single assay; seronegative non-responders in Class 3 might have antibodies detected using other assays or other

target antigens. We did not measure neutralizing antibodies or T-cell responses; however, neutralizing antibody responses are strongly correlated (Spearman's $\rho = 0.87$) with anti-spike binding antibodies following infection as previously reported[49]. This community survey had visits scheduled independent of infection or symptom status, so we could not precisely identify the start of infection or symptom onset; we therefore also incorporated positives from the national testing programme (targeting symptomatic infections) and used the first swab positive test and latent class models to indirectly estimate the start of infection. Similarly, we were not able to model antibody trajectories from each participant's maximum levels, as antibody data were collected monthly. However, we chose a starting point that was close to but slightly after the peak IgG level; although this could slightly underestimate peak IgG levels, the half-life will be unbiasedly estimated if the assumption of exponential decline is correct. Re-infections were rare, with only 92 (0.5%) participants with antibody data having potential re-infections > 120 days after their first infection episode (Supplementary Fig. 1). Most had only one antibody result, so it was impossible to investigate any boosting of antibody levels following re-infection.

In conclusion, in this representative study of infected individuals from the UK general population, around 1 in 4 people did not develop anti-spike IgG antibodies following a positive PCR test in regular screening. Non-responders were more likely to be older and not report symptoms. Among participants who seroconvert, anti-spike IgG antibodies remained above the positivity threshold for an average of 380–590 days for 20-year-olds, 410–649 days for 40-year-olds, 441–703 days for 60-year-olds and 471–755 days for 80-year-olds. These estimates of the durability of natural immunity may aid planning of the vaccination strategies. Further studies are required to determine the extent to which waning antibody levels impact immunity and protection following infection and vaccination and to assess the risk of infection in seronegative non-responders.

## Methods

**Population and settings.** The UK's ONS CIS (ISRCTN21086382) randomly selects private households on a continuous basis from address lists and previous surveys, to provide a representative sample across the UK's four countries (England, Wales, Northern Ireland and Scotland). After obtaining verbal agreement to participate, a study worker visited each household to take written informed consent from individuals ≥2 years. This consent was obtained from participants ≥16 years, parents/carers for those 2–15 years, whereas those 10–15 years also provided written assent. Children aged <2 years were not eligible for the study.

At the first visit, participants were asked for (optional) consent for follow-up visits every week for the next month, then monthly for 12 months from enrolment. Individuals were surveyed on their socio-demographic characteristics, behaviours, and vaccination status. Combined nose and throat swabs were taken from all consenting household members for SARS-CoV-2 PCR testing. Following written and verbal instruction, participants swabbed the back of their own throat, followed by both nostrils using the same swab (https://www.ndm.ox.ac.uk/files/coronavirus/covid-19-infection-survey/1510cisswabinstructionguideenglish_p.pdf).

For a random 10–20% of households, individuals ≥16 years were invited to provide blood samples monthly for serological testing. Participants with a positive swab test and their household members were also invited to provide blood monthly for follow-up visits. Details on the sampling design are provided elsewhere[36]. From April 2021, additional participants were invited to provide blood samples monthly to assess vaccine responses, based on a combination of random selection and prioritization of those in the study for the longest period (independent of test results). The study protocol is available at https://www.ndm.ox.ac.uk/covid-19/covid-19-infection-survey/protocol-and-information-sheets. The study received ethical approval from the South Central Berkshire B Research Ethics Committee (20/SC/0195).

**Laboratory testing.** Combined nose and throat swabs were tested at high-throughput national "Lighthouse" laboratories in Glasgow (from 16 August 2020 to present) and Milton Keynes (from 26 April 2020 to 8 February 2021). The presence of three SARS-CoV-2 genes (ORF1ab, nucleocapsid protein (N), and spike protein (S)) was identified using real-time PCR with the TaqPath RT-PCR COVID-19 kit (Thermo Fisher Scientific). PCR outputs were analysed using UgenTec Fast Finder 3.300.5 (TaqMan 2019-nCoV Assay Kit V2 UK NHS ABI 7500 v2.1; UgenTec),

with an assay-specific algorithm and decision mechanism that allows conversion of amplification assay raw data into test results with minimal manual intervention. Samples were called positive if at least a single $N$ and/or $ORF1ab$ gene were detected, and PCR traces exhibited an appropriate morphology. The $S$ gene alone is not considered to be a reliable positive[36].

SARS-CoV-2 antibody levels were tested on venous or capillary blood samples using an enzyme-linked immunosorbent assay (ELISA) detecting anti-trimeric spike IgG developed by the University of Oxford[30,36]. Normalized results are reported in ng ml$^{-1}$ of mAb45 monoclonal antibody equivalents. Before 26 February 2021, the assay used fluorescence detection as previously described, with a positivity threshold of 8 million units validated on banks of known SARS-CoV-2-positive and -negative samples[30]. After this, it used a commercialized CE-marked version of the assay, the Thermo Fisher OmniPATH 384 Combi SARS-CoV-2 IgG ELISA (Thermo Fisher Scientific), with the same antigen and colorimetric detection. mAb45 is the manufacturer-provided monoclonal antibody calibrant for this quantitative assay. To allow conversion of fluorometrically determined values in arbitrary units, we compared 3840 samples, which were run in parallel on both systems. A piece-wise linear regression was used to generate the following conversion formula:

$$\log_{10}(mAb45\ units) = 0.221738 + 1.751889e - 07 * \text{fluorescence\_units} +$$
$$5.416675e - 07 * (\text{fluorescence\_units} > 9190310) * (\text{fluorescence\_units} - 9190310)$$
$$(1)$$

We used 42 ng ml$^{-1}$ as the threshold for an IgG-positive or -negative result (corresponding to the 8 million units with fluorescence detection). We also analysed results using two alternative thresholds: first, 28 ng ml$^{-1}$ (~7 million fluorescence units), which we had previously found, corresponded to 50% protection against any asymptomatic/symptomatic re-infection following a previous infection[4]. We also used 6 ng ml$^{-1}$, the level expected to correspond to 50% protection against severe infection, on the basis of this level of protection being associated with neutralizing antibody levels at 3% of peak levels in a previous report[25]. Given the lower and upper limits of the assay, measurements <2 ng ml$^{-1}$ (46 observations, 0.3%) and >800 ng ml$^{-1}$ (259 observations, 1.8%) were truncated at 2 and 800 ng ml$^{-1}$, respectively.

Each batch of 320 samples (diluted 1 : 50) was run with a negative control sample (Sigma human serum H6194, diluted 1 : 50) run in duplicate and dilution series of 3 monoclonal antibodies[50–52] run in duplicate used for assay calibration and quality control (CR3022: 4 dilutions [1000, 300, 100, 30 ng ml$^{-1}$], mAb45: 5 dilutions [400, 300, 100, 30, 10 ng ml$^{-1}$], mAb269: 4 dilutions [300, 100, 30, 10 ng ml$^{-1}$]). Values obtained for each control and calibration sample were compared to established historic control values and plates subjected to acceptance criteria that required all 28 controls and calibrants to fall within historic limits, namely no more than 5 control samples >2 SDs different, no more than 2 samples >3 SDs different and no more than 1 sample >4 SDs different. The first two limits were based on rejecting batches where the probability of the observed variation excluded that expected 99% of the time. The latter rule allowed for one-off robotic error.

**Statistical analysis**. This analysis included participants aged 16 years and over, who had SARS-Cov-2 infection (defined by a positive PCR test) from 26 April 2020 to 14 June 2021. As multiple positive swab tests could be obtained at follow-up visits, positive PCR tests were grouped into 'episodes'. We used the first episode (starting with the first positive PCR or index positive) for each participant in the main analysis. Second episodes, defined by a repeat PCR positive >120 days after the start of the first infection (associated with risk reductions for new positive episodes of similar magnitude to vaccination[53]) were excluded.

Study visits occur on a fixed schedule, meaning that infection episodes could be identified up to 30 days or more after onset (as well as 'early' in some pre-symptomatic cases). As participants were told to obtain a test from the national testing programme if symptomatic, to improve our estimate of the start of each infection episode, we linked study data to data on swab positivity from the English national testing programme (data were not available for Scotland, Wales and Northern Ireland). The national testing programme is intended for individuals with symptoms (although a substantial proportion report no symptoms) and so not all PCR-positive episodes in the English study participants also have a positive test from the national testing programme. For this analysis, we used the date of the first positive PCR test in the study or the national testing programme as the start of the episode, whichever came first. Ct values and gene positivity patterns are not available from the national testing programme and so these factors were obtained from PCR-positive samples in the ONS survey only.

We included all antibody measurements from 90 days before each participant's first swab positive date (index positive) through to 180 days after (~95 percentile), to avoid undue influence from outliers at late time points. We also excluded all antibody measurements taken from 3 days after the first vaccination. Vaccination status was self-reported at study visits and also linked to the National Immunisation Management Service (NIMS) in England, which contains all individuals' vaccination data in the English National Health Service COVID-19 vaccination programme. There was good agreement between self-reported and administrative vaccination data (98% on type and 95% on date[53]). We used

vaccination data from NIMS where available, for participants from England, and otherwise data from the survey.

We used the Ct value as the proxy of viral burden, defined as the minimum from all positive swab tests in the infection episode and categorizing at <30 to indicate moderate to higher viral burden. This threshold is used in the UK in algorithms for review of low-level positives at the laboratories where the PCR tests were performed and as a threshold for attempting whole-genome sequencing[53]. Gene positivity pattern during the episode was classified as three groups: (1) a single $ORF1ab$ gene or a single $N$ gene positive; (2) Alpha (B.1.1.7) SARS-CoV-2 variant compatible (at least once positive for $ORF1ab + N$ across the episode and never $S$ positive); and (3) $S$-positive ($ORF1ab + N + S$ or $ORF1ab + S$ or $N + S$ at least once across the episode). Participants with missing information on Ct values or gene positivity patterns or symptoms in the episode were excluded from analysis ($N = 133$). Self-reported symptoms were those reported at any visit within 35 days after the index positive date or reported to the national testing programme. Fever, cough, loss of smell and loss of taste were considered 'classic symptoms'.

We first used latent class mixed models (LCMMs) to identify distinct patterns of antibody response after natural infection, counting the date of the index positive in the survey as time 0. Restricted natural cubic splines (internal knots at −10, 30 and 60 days, and boundary knots at −60 and 140 days) were used to model time since the index positive as the fixed effect. A random-effect intercept and random-effect slope on all time spline variables were added to account for individual variability. The location of the knots was chosen to reflect fitted antibody trajectories in models with greater numbers of knots, which would not converge while also allowing for random effects. Age as a natural cubic spline (internal knots at 50 years and boundary knots at 20 and 80 years), presence of self-reported long-term health conditions, Ct value and self-reported symptoms were included as covariates for class membership[54]. The number of classes, up to a maximum of 4, was determined by examining and comparing the shape of the class trajectories and measures of model fit using Bayesian information criterion.

We used Bayesian linear mixed interval censored models to estimate the decay in antibody responses from their peak level, excluding those who did not seroconvert, and any participant with a positive or equivocal antibody result strictly before their index positive date (≥23 ng ml$^{-1}$) ($N = 6$) or a negative antibody measurement within 42 days of their first index positive ($N = 13$) (Supplementary Fig. 1). Time zero (peak level) for this analysis was determined from the estimated trajectories for each class from the LCMM (see 'Results'). We initially assumed an exponential fall in antibody levels over time, i.e., a linear decline on a $\log_2$ scale. This exponential decline model had been widely used in studies on antibody kinetics[17,24,25]. Population-level fixed effects, individual-level random effects for intercept and slope, and covariance between random effects were included in the model. The outcome was right-censored at 800 reflecting truncation of IgG values at 800 ng ml$^{-1}$. We excluded a very small number of measurements ($n = 24$) below 23 ng ml$^{-1}$ (likely reflecting mislabelled samples) to reduce the influence of outliers (Fig. 1). We examined the association between peak levels and antibody half-lives with age, sex, ethnicity, reporting having long-term health conditions, Ct values and self-reported symptoms. As sensitivity analysis, we also included a time-dependent variable to reflect pre-Alpha (before 16 Nov 2020) and Alpha (17 Nov 2020–16 May 2021) periods. Due to the short follow-up time when the Delta variant was dominant in our study (17 May 2021 onwards), there were no Delta period participants in the model. We tested for evidence of non-linearity in antibody declines on the log scale using two alterative models, a piece-wise linear regression allowing for a biphasic exponential decline with a knot at 28 days post peak and a model using natural cubic splines to allow for a more flexible non-linear fit. Model fits were compared using the leave-one-out cross-validation information criterion. Although the spline-based model provided a slightly better fit (Supplementary Fig. 10), extrapolating cubic polynomial models can lead to unstable estimates, so we used the biphasic model to predict times above different thresholds.

For each Bayesian linear mixed interval censored model, weakly informative priors were used (Supplementary Table 9). Four chains were run per model with 4000 iterations and a warm-up period of 2000 iterations to ensure convergence, which was confirmed visually and by ensuring the Gelman–Rubin statistic was <1.05 (Supplementary Table 6). Then, 95% CrIs were calculated using highest posterior density intervals.

As sensitivity analyses, we additionally used 400 and 500 as the censoring threshold for IgG levels and chose different starting points to examine robustness.

Data preparation was conducted using Stata MP16. All analyses were performed in R 3.6 using the following packages: tidyverse (version 1.3.0), brms (version 2.14.0), rstanarm (version 2.21.1), splines (version 3.6.1), lcmm (version 1.9.2), nnet (version 7.3-14), ggeffects (version 0.14.3), arsenal (version 3.4.0), cowplot (version 1.1.0) and bayesplot (version 1.7.2).

**Reporting summary**. Further information on research design is available in the Nature Research Reporting Summary linked to this article.

## Data availability

De-identified study data are available for access by accredited researchers in the ONS Secure Research Service (SRS) for accredited research purposes under part 5, chapter 5 of the Digital Economy Act 2017. Individuals can apply to be an accredited researcher using

the short form on https://researchaccreditationservice.ons.gov.uk/ons/ONS_registration.ofml. Accreditation requires completion of a short free course on accessing the SRS. To request access to data in the SRS, researchers must submit a research project application for accreditation in the Research Accreditation Service (RAS). Research project applications are considered by the project team and the Research Accreditation Panel (RAP) established by the UK Statistics Authority at regular meetings. Project application example guidance and an exemplar of a research project application are available. A complete record of accredited researchers and their projects is published on the UK Statistics Authority website to ensure transparency of access to research data. For further information about accreditation, contact Research.Support@ons.gov.uk or visit the SRS website.

## Code availability

A copy of the analysis code is available at https://github.com/jiaweioxford/COVID19_infection_antibody_response (https://doi.org/10.5281/zenodo.5541764).

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

## Acknowledgements
We are grateful for the support of all COVID-19 Infection Survey participants. This study is funded by the Department of Health and Social Care with in-kind support from the Welsh Government, the Department of Health on behalf of the Northern Ireland Government and the Scottish Government. J.W. is supported by University of Oxford and the China Scholarship Council. A.S.W., T.E.A.P., N.S., D.E. and K.B.P. are supported by the National Institute for Health Research Health Protection Research Unit (NIHR HPRU) in Healthcare Associated Infections and Antimicrobial Resistance at the University of Oxford in partnership with Public Health England (PHE) (NIHR200915). A.S.W. and T.E.A.P. are also supported by the NIHR Oxford Biomedical Research Centre. K.B.P. is also supported by the Huo Family Foundation. A.S.W. is also supported by core support from the Medical Research Council UK to the MRC Clinical Trials Unit [MC_UU_12023/22] and is an NIHR Senior Investigator. P.C.M. is funded by Wellcome (intermediate fellowship, grant ref 110110/Z/15/Z) and holds an NIHR Oxford BRC Senior Fellowship award. D.W.E. is supported by a Robertson Fellowship and an NIHR Oxford BRC Senior Fellowship. NS is an Oxford Martin Fellow and holds an NIHR Oxford BRC Senior Fellowship. The views expressed are those of the authors and not necessarily those of the National Health Service, NIHR, Department of Health, or PHE.

## Author contributions
The study was designed and planned by A.S.W., J.F., J.B., J.N., I.D. and K.B.P., and is being conducted by A.S.W., R.S., E.R., A.H., B.M., T.E.A.P., P.C.M., N.S., S.H., E.Y.J., D.I.S., D.W.C. and D.W.E. This specific analysis was designed by J.W., D.W.E., A.S.W. and K.B.P. J.W., K.B.P., T.M. and L.L. contributed to the statistical analysis of the survey data. J.W., D.W.E., K.B.P. and A.S.W. drafted the manuscript and all authors contributed to interpretation of the data and results, and revised the manuscript. All authors approved the final version of the manuscript.

## Competing interests
D.W.E. declares lecture fees from Gilead, outside the submitted work. No other author has a conflict of interest to declare. For the purpose of Open Access, the author has applied a CC BY public copyright licence to any Author Accepted Manuscript version arising from this submission.

## Additional information

## the COVID-19 Infection Survey team

Tina Thomas[6], Duncan Cook[6], Daniel Ayoubkhani[6], Russell Black[6], Antonio Felton[6], Megan Crees[6], Joel Jones[6], Lina Lloyd[6], Esther Sutherland[6], Emma Pritchard[1], Karina-Doris Vihta[1], George Doherty[1], James Kavanagh[1], Kevin K. Chau[1], Stephanie B. Hatch[1], Daniel Ebner[1], Lucas Martins Ferreira[1], Thomas Christott[1], Wanwisa Dejnirattisai[1], Juthathip Mongkolsapaya[1], Sarah Cameron[1], Phoebe Tamblin-Hopper[1], Magda Wolna[1], Rachael Brown[1], Richard Cornall[1], Gavin Screaton[1], Katrina Lythgoe[1], David Bonsall[1], Tanya Golubchik[1], Helen Fryer[1], Stuart Cox[13], Kevin Paddon[13], Tim James[13], Thomas House[14], Julie Robotham[8], Paul Birrell[8], Helena Jordan[15], Tim Sheppard[15], Graham Athey[15], Dan Moody[15], Leigh Curry[15], Pamela Brereton[15], Ian Jarvis[16], Anna Godsmark[16], George Morris[16], Bobby Mallick[16], Phil Eeles[16], Jodie Hay[17], Harper VanSteenhouse[17], Jessica Lee[18], Sean White[19], Tim Evans[19], Lisa Bloemberg[19], Katie Allison[20], Anouska Pandya[20], Sophie Davis[20], David I. Conway[21], Margaret MacLeod[21] & Chris Cunningham[21]

[13]Oxford University Hospitals NHS Foundation Trust, Oxford, UK. [14]University of Manchester, Manchester, UK. [15]IQVIA, London, UK. [16]National Biocentre, Milton Keynes, UK. [17]Glasgow Lighthouse Laboratory, London, UK. [18]Department of Health and Social Care, London, UK. [19]Welsh Government, Cardiff, UK. [20]Scottish Government, Edinburgh, UK. [21]Public Health Scotland, Edinburgh, UK.

