## [Peer Review File · Nature Communications]

Anti-spike antibody response to natural SARS-CoV-2 infection in the general populationReviewers' Comments:

Reviewer #1:

Remarks to the Author:

My name is Enrico Lavezzo and I prefer non-anonymous reviews to put reviewer and authors on the same "peer" level.

Thank you for the opportunity to review "Anti-spike antibody response to natural SARS-CoV-2 infection in the general population". This is a very informative study based on a big dataset of thousands of patients who recovered from COVID-19 and coming from Office for National Statistics (ONS) COVID-19 Infection Survey (CIS), UK. The authors exploited a combination of swab and serological data to draw important conclusions about the prevalence of non-responders (in terms of anti-SARS-CoV-2 antibody production), the predictors of missing responses (mainly higher Ct threshold, used as a proxy for the viral load, the absence of self-reported symptoms, older age and not working with patients), and the determinants of antibody peak level and half-life (ethnicity and Ct, ethnicity and sex, respectively). Finally, they provide estimates about the duration of antibodies and the level of protection they confer, also taking into consideration viral evolution and the effect of delta VOC.

I did not find major issues throughout the manuscript, which reads well and provides timely discussions on some counter-intuitive results that emerged. I found particularly elegant the approach to identify post-infection classes based on a latent class model.

I report some minor points which I think the authors should address to make the paper even more complete and informative:

- I found few information in the text about the swab sampling method. Since it is known from the literature that different sampling techniques (e.g. nasopharyngeal and oropharyngeal swabs) have different sensitivities, specificities and, as a consequence, influence the Ct values (see Wang et al. below, which is just one of papers about this topic but many other sources of variation have been considered, including whether the nasopharynx is swabbed through just one or both nostrils), I wonder whether this could have impacted the results of this analysis. I feel the authors should discuss this potential issue.

Wang X, Tan L, Wang X, et al. Comparison of nasopharyngeal and oropharyngeal swabs for SARS-CoV-2 detection in 353 patients received tests with both specimens simultaneously. *Int J Infect Dis.* 2020;94:107-109. doi:10.1016/j.ijid.2020.04.023

- Figure S4: this figure shows the probability of being non-responsive according to age, it is not related to specific symptoms as I would expect from the sentence in the main text (beginning of page 7)

- End of page 6: mostly strongly \diamond more strongly?

Overall this is a nice paper with a lot of effort, I look forward to seeing this in print.

P.S. If the authors have any questions that they feel can expedite the revision process if discussed back & forth, they are free to contact me at [email address redacted from Peer Review File]

Reviewer #2:

Remarks to the Author:

In the considered manuscript entitled "Anti-spike antibody response to natural SARS-CoV-2 infection in the general population", Wei and colleagues investigate the highly relevant question of trajectory and duration of anti-SARS-CoV-2 antibodies and predictors of seroconversion. This is a particularly

relevant investigation in the U.K., as well as the U.S., where vaccine hesitancy is heightened and where uptake has stubbornly stalled, particularly with the recent discourse on the need and timing of booster vaccinations with consideration for global vaccine fairness.

They leverage ONS data to report the number of “non-responders”, predictors thereof, and model an estimated half-life of 184 days with a projected duration of protection estimated to be 1.5 to 2 years from time of seroconversion.

Major comments:

-At several points in the paper, but most glaringly in the initial presentation of the three classes of post-infection, there is an over-reliance on highly significant p-values from univariate group testing in a population-scale study. Describing the differences by class would be more appropriate, e.g., instead of “a higher percentage of reported symptoms (77.7%, $p < 0.001$ vs. any other class), reporting the proportions of each class would be more informative.

-I am a bit worried that some of the more compelling pieces of the study, the sections entitled, “Determinants of the peak and half-life of antibody responses” and “Duration of antibody responses and possible associated immune protection” are based on an assumption of exponentially decreasing antibody levels based on peak IgG levels in a subset of patients, rather than more empirical/longitudinally sampled data. A significant expansion of the text justifying this approach--and its use in other contexts--would greatly help readers such as myself, who might be unfamiliar with infectious disease epidemiology/dynamics and could be reasonably expected in an audience as broad as Nature Communications.

-Given the long period of study (for this pandemic, April 2020 to June 2021) as well as the significant impact of symptoms on class determination, more should be done to address the evolving natural history of the disease, especially in the context of both the delta variant and the early successes of the mass vaccination campaign, which surely contributed to prevalence of overt symptoms. Multivariable modeling inclusive of time period or national prevalence vs. personal vaccine status would be important to consider.

Minor comments:

-At the beginning of page 5, many concepts are introduced without context (Ct, ORF1ab/N, etc.) and the manuscript should be revised to account for the sequential fashion in which the study would be read, if accepted (e.g., Intro to Results).

-Given the problems that were acknowledged with class 2, it may be appropriate to relegate most of their findings to a Supplementary Appendix.

-The Discussion really strengthens towards the end but some of the earlier text is a bit duplicative (beyond the recitation of findings, which is to be expected) and could probably be streamlined.

-Multiple figures with wide y-axis ranges would benefit from being plotted on a log scale, while Table 2 could be presented as a forest plot.

Reviewer #3:

Remarks to the Author:

This is a terrific analysis of data from a very large study of SARS-CoV-2 infection in the UK population. Collected samples are processed using a molecular PCR (3 gene) assay and a serological (anti-Spike IgG) assay. In my opinion, the most interesting result is on the 3 very different categories of serological response following a positive PCR test (Figure 1). In particular, the finding that 24% of individuals with SARS-CoV-2 infection (positive PCR swab) do not generate a detectable antibody response, has potentially very important consequences. As this finding may be interpreted as “one quarter of people don’t generate immunity after SARS-CoV-2 infection” it needs to be treated with caution.

I would want to see a more careful interpretation of the finding in terms of:

- (1) It really is the case that 24% don't generate an antibody response
- (2) High rates of false positive PCRs
- (3) Low rates of false negative serological tests (especially in light of the previously reported 99% sensitivity of the assay).

I appreciate the authors have already partially addressed these points - I very much liked the sensitivity analysis of Ct < 32 and 2 gene positivity. However, I would want to see the more conservative position that the 24% figure is due to a combination of these factors.

Serological cutoffs

There are 3 anti-Spike IgG cutoffs that are critical to this analysis.

- IgG positivity = 42 ng/mL (\leq 8 million AU)
- 50% protection against symptomatic infection = 28 ng/mL (\leq 7 million AU)
- 50% protection against severe infection = 6 ng/mL

The IgG positivity threshold is based on an analysis by Ainsworth et al based on a comparison of 536 positive samples and 976 pre-pandemic negative samples. The reported specificity was 99.0% (95% CI: 98.1%, 99.5%), and the reported sensitivity was 99.4% (95% CI: 98.2%, 99.9%). Based on the data used to validate this cutoff, one would expect 0.6% non-responders. This is substantially different to the 24% rate observed in the present study, indicating that there must be important differences between the samples used for assay validation and the present samples (presumably due to the presence of greater symptoms in the validation samples used by Ainsworth et al).

The cutoff for protection is based on an analysis by Lumley et al. Notably this cutoff is lower than the cutoff for IgG positivity. Therefore, there will be individuals who have no detectable Spike IgG (according to the definition <42 ng/mL) but who are protected.

For the cutoff for 50% protection against severe infection, the authors reasoning is clear (3% of an average peak of 200 ng/mL), but there are some large uncertainties that need to be acknowledged. Firstly, the 3% from Khoury et al is an estimate with its own uncertainty. Secondly, this estimate is based on neutralizing titre, and the authors estimate is based on anti-Spike IgG. Although there is a clear correlation between neutralizing titre and anti-Spike IgG this is again uncertainty in this relationship which should be acknowledged.

There are two things which I would consider important:

Necessary: validate anti-Spike IgG samples on negative control samples from the same study

Optional: test samples using an anti-N IgG assay

Kinetics

The authors' estimates of duration of protection are based on an assumption of exponential decay. In Figures 1 and S3, the spline fit predicts a different pattern of boosting, followed by rapid decay, followed by slower decay. In Figure S10b, we see a really a reasonably good agreement between the spline and linear fits within the time since infection of 120 days. Within this time range I'm certainly very happy with the linear approximation. The challenge is when we extrapolate into the future beyond the range of the data (the y-axis in Figure 4 goes up to 2100 days). We can already see the linear and spline models starting to diverge in Figure S10b at time = 120 days.

There is a growing body of evidence to indicate that anti-Spike IgG patterns wane according to a bi-phasic exponential and not an exponential pattern over longer time periods (Wheatley et al, and Pelleau et al). Indeed, we can see the same bi-phasic exponential pattern in Figure 1 and Figure 3a.

The authors should consider fitting a bi-phasic exponential instead of an exponential model.

Minor point

I think figures of the style Figure S3 are much clearer and more informative than Figure 1. The authors could consider replacing Figure 1 with Figure S3.

References

Ainsworth, M. et al. Performance characteristics of five immunoassays for SARS-CoV-2: a head-to-head benchmark comparison. *The Lancet Infectious Diseases* 20, 1390–1400 (2020).

Lumley, S. F. et al. Antibody Status and Incidence of SARS-CoV-2 Infection in Health Care Workers. *New England Journal of Medicine* 384, 533–540 (2021).

Wheatley et al. Evolution of immune responses to SARS-CoV-2 in mild-moderate COVID-19. *Nature Comms.* 2021;

Pelleau et al. Kinetics of the SARS-CoV-2 antibody response and serological estimation of time since infection. *J Inf Dis.* 2021;

REVIEWER COMMENTS

Reviewer #1 (Remarks to the Author):

My name is Enrico Lavezzo and I prefer non-anonymous reviews to put reviewer and authors on the same “peer” level.

Thank you for the opportunity to review “Anti-spike antibody response to natural SARS-CoV-2 infection in the general population”. This is a very informative study based on a big dataset of thousands of patients who recovered from COVID-19 and coming from Office for National Statistics (ONS) COVID-19 Infection Survey (CIS), UK. The authors exploited a combination of swab and serological data to draw important conclusions about the prevalence of non-responders (in terms of anti-SARS-CoV-2 antibody production), the predictors of missing responses (mainly higher Ct threshold, used as a proxy for the viral load, the absence of self-reported symptoms, older age and not working with patients), and the determinants of antibody peak level and half-life (ethnicity and Ct, ethnicity and sex, respectively). Finally, they provide estimates about the duration of antibodies and the level of protection they confer, also taking into consideration viral evolution and the effect of delta VOC.

I did not find major issues throughout the manuscript, which reads well and provides timely discussions on some counter-intuitive results that emerged. I found particularly elegant the approach to identify post-infection classes based on a latent class model.

I report some minor points which I think the authors should address to make the paper even more complete and informative:

- I found few information in the text about the swab sampling method. Since it is known from the literature that different sampling techniques (e.g. nasopharyngeal and oropharyngeal swabs) have different sensitivities, specificities and, as a consequence, influence the Ct values (see Wang et al. below, which is just one of papers about this topic but many other sources of variation have been considered, including whether the nasopharynx is swabbed through just one or both nostrils), I wonder whether this could have impacted the results of this analysis. I feel the authors should discuss this potential issue.

Wang X, Tan L, Wang X, et al. Comparison of nasopharyngeal and oropharyngeal swabs for SARS-CoV-2 detection in 353 patients received tests with both specimens simultaneously. *Int J Infect Dis.* 2020;94:107-109. doi:10.1016/j.ijid.2020.04.023

Response: We have added to the methods that participants performed their own swabs of the throat and both nostrils (one swab) – the main reason was to ensure that home visits were “non-contact” with the study workers staying >2m away throughout to avoid transmission risk in both directions. We have also added a link to the written instructions provided to participants (page 13).

We have added to the discussion that swabbing technique may have affected the sensitivity of PCR testing, however prior infection could also be detected serologically, and so the overall impact of this on the findings is likely to be small.

- **Figure S4: this figure shows the probability of being non-responsive according to age, it is not related to specific symptoms as I would expect from the sentence in the main text (beginning of page 7)**

Response: Figure S4 is shown with variables other than age set to reference categories as described in the legend. New Figure 3 and Table S3 shows the coefficients from the symptom variables which are binary variables. Since we fitted a spline to the continuous age, we did not present the

coefficients in new Figure 3 and Table S3 as these are not readily interpretable as numbers, instead we plotted the relationship in Figure S4. So, Figure S4 is an extension to Figure 3 and Table S3. We have clarified this where we reference the Figure in the text: "(Fig. 3, Supplementary Table 3, with non-linear effect of age in Supplementary Fig. 4)" (page 7).

- End of page 6: mostly strongly \diamond more strongly?

Response: We have changed "mostly" to "most".

Overall this is a nice paper with a lot of effort, I look forward to seeing this in print.

P.S. If the authors have any questions that they feel can expedite the revision process if discussed back & forth, they are free to contact me at [email address redacted from Peer Review File]

Reviewer #2 (Remarks to the Author):

In the considered manuscript entitled "Anti-spike antibody response to natural SARS-CoV-2 infection in the general population", Wei and colleagues investigate the highly relevant question of trajectory and duration of anti-SARS-CoV-2 antibodies and predictors of seroconversion. This is a particularly relevant investigation in the U.K., as well as the U.S., where vaccine hesitancy is heightened and where uptake has stubbornly stalled, particularly with the recent discourse on the need and timing of booster vaccinations with consideration for global vaccine fairness.

They leverage ONS data to report the number of "non-responders", predictors thereof, and model an estimated half-life of 184 days with a projected duration of protection estimated to be 1.5 to 2 years from time of seroconversion.

Major comments:

-At several points in the paper, but most glaringly in the initial presentation of the three classes of post-infection, there is an over-reliance on highly significant p-values from univariate group testing in a population-scale study. Describing the differences by class would be more appropriate, e.g., instead of "a higher percentage of reported symptoms (77.7%, $p < 0.001$ vs. any other class), reporting the proportions of each class would be more informative.

Response: We now set out the three classes at the start of the "Antibody trajectories following SARS-CoV-2 infection" section, and report median [IQR] or proportions for comparisons without p-values as suggested (page 5-6).

-I am a bit worried that some of the more compelling pieces of the study, the sections entitled, "Determinants of the peak and half-life of antibody responses" and "Duration of antibody responses and possible associated immune protection" are based on an assumption of exponentially decreasing antibody levels based on peak IgG levels in a subset of patients, rather than more empirical/longitudinally sampled data. A significant expansion of the text justifying this approach--and its use in other contexts--would greatly help readers such as myself, who might be unfamiliar with infectious disease epidemiology/dynamics and could be reasonably expected in an audience as broad as Nature Communications.

Response: The spaghetti plots that have been moved to Figure 1 illustrate the challenge with trying to fit one model to all the empirical data – different participants have different underlying trajectories and modelling them all together removes the ability to identify the key features of each. The latent class models however are an empirical approach to identifying which participant belongs to which class, so the sub-populations are identified by the data. The “classical seroconverters” are the expected trajectory that Class 2 would also have followed, but we simply do not know the time of the initial infection well enough to model them together. We have changed the first sentence in the paragraph describing peak and half-lives to clarify this (page 7). We also added a few references supporting using the exponential model to estimate antibody levels.

In terms of the specific functional form for the association between antibody levels (on the log scale) and time from infection, we previously tested the exponential assumption by visually comparing the decline in a spline-based model and a linear model on the log scale (i.e. an exponential decline). These two models yielded very similar trajectories supporting the exponential assumption (Figure S10, Figure 4).

To further test the exponential assumption we have added additional analyses. We additionally present a piecewise linear model on the log scale, i.e. a bi-exponential decline. We then compare the three models (linear, piecewise linear, spline-based) using the leave-one-out cross-validation information criterion (LOOIC) using the `loo()` and `loo_compare()` functions in `rstanarm`. We found that the spline model has the lowest LOOIC (11402), followed by the bi-exponential model (11594), then the linear model (11819). The LOOIC difference is larger than $2 \times SE$ (standard error), suggesting the spline model has a better fit.

However, over the first 120 days after infection all 3 models have similar trajectories (Figure 4, S10), therefore we have retained the linear model for the investigation of the covariates associated with antibody levels presented in new Table 2, as this yields the most interpretable coefficients. We have added descriptions and references in the Methods section on the approaches.

For predictions of the duration of response, given the better fit of the non-linear model, we present findings using the bi-exponential model as compared with the spline model this is less susceptible to model instability in the tails of the fitted values.

-Given the long period of study (for this pandemic, April 2020 to June 2021) as well as the significant impact of symptoms on class determination, more should be done to address the evolving natural history of the disease, especially in the context of both the delta variant and the early successes of the mass vaccination campaign, which surely contributed to prevalence of overt symptoms. Multivariable modeling inclusive of time period or national prevalence vs. personal vaccine status would be important to consider.

Response: As stated in the first paragraph of Results, we only considered antibody responses in those who were not vaccinated, so the vaccination programme would not impact on this analysis. We have previously reported separately on antibody responses following vaccination (<https://www.nature.com/articles/s41564-021-00947-3>).

We agree that different time periods and dominant variants may have impacted findings. We therefore performed a sensitivity analysis where we added another variable to reflect pre-Alpha and Alpha periods in the Bayesian linear mixed model. Due to the short follow-up time when the Delta variant was dominant in our study (17 May 2021 onwards), there were in fact no Delta period participants in the Bayesian model. We did not find significant effects from different time periods on antibody responses, thus we kept the original model without time period variable as our main results. However, we report the results of the sensitivity analyses in the results (Table S7) and have added them to the methods text.

Minor comments:

-At the beginning of page 5, many concepts are introduced without context (Ct, ORF1ab/N, etc.) and the manuscript should be revised to account for the sequential fashion in which the study would be read, if accepted (e.g., Intro to Results).

Response: We have added a sentence before the introduction of SARS-CoV-2 genes in page 5 as suggested. The concept of Ct was introduced in the last paragraph of the Introduction: "PCR cycle threshold (Ct) values (inversely related to viral load)".

-Given the problems that were acknowledged with class 2, it may be appropriate to relegate most of their findings to a Supplementary Appendix.

Response: We currently only compare class 1 to class 3 when considering non-response, and so there is relatively little existing text on class 2. We would prefer to retain this when it is first described to help readers understand why it is present, and also so that the main Results does reflect the totality of the data, since, as noted above, other analyses focus on the different sub-populations.

-The Discussion really strengthens towards the end but some of the earlier text is a bit duplicative (beyond the recitation of findings, which is to be expected) and could probably be streamlined.

Response: We have attempted to streamline the first half of the discussion as suggested, which also makes space to add additions suggested by reviewers.

-Multiple figures with wide y-axis ranges would benefit from being plotted on a log scale, while Table 2 could be presented as a forest plot.

Response: We have modified old Table 2 using a forest plot (new Figure 3). Figures 1 and 4 are presented with the y-axis on a log₁₀ scale. We prefer to leave Figure 5 on a linear scale to aid reading specific values off the axis.

Reviewer #3 (Remarks to the Author):

This is a terrific analysis of data from a very large study of SARS-CoV-2 infection in the UK population. Collected samples are processed using a molecular PCR (3 gene) assay and a serological (anti-Spike IgG) assay. In my opinion, the most interesting result is on the 3 very different categories of serological response following a positive PCR test (Figure 1). In particular, the finding that 24% of individuals with SARS-CoV-2 infection (positive PCR swab) do not generate a detectable antibody response, has potentially very important consequences. As this finding may be interpreted as "one quarter of people don't generate immunity after SARS-CoV-2 infection" it needs to be treated with caution.

I would want to see a more careful interpretation of the finding in terms of:

- (1) It really is the case that 24% don't generate an antibody response
- (2) High rates of false positive PCRs
- (3) Low rates of false negative serological tests (especially in light of the previously reported 99% sensitivity of the assay).

I appreciate the authors have already partially addressed these points - I very much liked the sensitivity analysis of Ct < 32 and 2 gene positivity. However, I would want to see the more conservative position that the 24% figure is due to a combination of these factors.

Response: We agree it is important to avoid the conclusion that “one quarter of people don’t generate immunity after SARS-CoV-2 infection” if this is not the case.

We have expanded the results and discussion to present the denominator for the number of participants with strong evidence of a PCR positive result (Ct ≤32, ≥2 genes positive), which yields a percentage of 11% non-response after such a PCR result.

We have added to the discussion the possibility of false negative serology results, and adding some text to reconcile the previously reported sensitivity of the antibody assay of 99% with the lower apparent sensitivity in this population.

We already cover the possibility of false positive PCR results in some detail, with the positive predictive value >95% meaning that even with imperfect performance most non-responders did have a true positive result, we have added to the text here too.

Serological cutoffs

There are 3 anti-Spike IgG cutoffs that are critical to this analysis.

- **IgG positivity = 42 ng/mL (<=> 8 million AU)**
- **50% protection against symptomatic infection = 28 ng/mL (<=> 7 million AU)**
- **50% protection against severe infection = 6 ng/mL**

The IgG positivity threshold is based on an analysis by Ainsworth et al based on a comparison of 536 positive samples and 976 pre-pandemic negative samples. The reported specificity was 99.0% (95% CI: 98.1%, 99.5%), and the reported sensitivity was 99.4% (95% CI: 98.2%, 99.9%). Based on the data used to validate this cutoff, one would expect 0.6% non-responders. This is substantially different to the 24% rate observed in the present study, indicating that there must be important differences between the samples used for assay validation and the present samples (presumably due to the presence of greater symptoms in the validation samples used by Ainsworth et al).

Response: We have added to our discussion possible explanations for these differences, which most likely arise from the different populations studied – unwell, symptomatic, often hospitalised patients in Ainsworth et al, compared to those with asymptomatic or predominantly mild infection in this study.

The cutoff for protection is based on an analysis by Lumley et al. Notably this cutoff is lower than the cutoff for IgG positivity. Therefore, there will be individuals who have no detectable Spike IgG (according to the definition <42 ng/mL) but who are protected.

Response: We agree, but Figure 1 shows that individuals with levels between 28 ng/ml and 42 ng/ml are relatively uncommon. We have added this point to the discussion.

For the cutoff for 50% protection against severe infection, the authors reasoning is clear (3% of an average peak of 200 ng/mL), but there are some large uncertainties that need to be acknowledged. Firstly, the 3% from Khoury et al is an estimate with its own uncertainty. Secondly, this estimate is based on neutralizing titre, and the authors estimate is based on anti-Spike IgG.

Although there is a clear correlation between neutralizing titre and anti-Spike IgG this is again uncertainty in this relationship which should be acknowledged.

Response: We have added these uncertainties to the discussion.

There are two things which I would consider important:

Necessary: validate anti-Spike IgG samples on negative control samples from the same study

Optional: test samples using an anti-N IgG assay

Response: We have added further detail to the methods to detail the robust use of control samples throughout the processing of samples:

“Each batch of 320 samples was run with a negative control sample (Sigma human serum H6194) run in duplicate and dilution series of 3 monoclonal antibodies run in duplicate used for assay calibration and quality control (CR3022 – 4 dilutions, mAb45 – 5 dilutions, mAb269 – 4 dilutions; the latter two produced at the University of Oxford). Values obtained for each control and calibration sample were compared to established historic control values and plates subjected to acceptance criteria that required all 28 controls and calibrants to fall within historic limits, namely, no more than 5 control samples >2 standard deviations different, no more than 2 samples >3 standard deviations different and no more than 1 sample >4 standard deviations different. The first two limits were based on rejecting batches where the probability of the observed variation excluded that expected 99% of the time. The latter rule allowed for one-off robotic error.”

Kinetics

The authors’ estimates of duration of protection are based on an assumption of exponential decay. In Figures 1 and S3, the spline fit predicts a different pattern of boosting, followed by rapid decay, followed by slower decay. In Figure S10b, we see a really a reasonably good agreement between the spline and linear fits within the time since infection of 120 days. Within this time range I’m certainly very happy with the linear approximation. The challenge is when we extrapolate into the future beyond the range of the data (the y-axis in Figure 4 goes up to 2100 days). We can already see the linear and spline models starting to diverge in Figure S10b at time = 120 days.

There is a growing body of evidence to indicate that anti-Spike IgG patterns wane according to a bi-phasic exponential and not an exponential pattern over longer time periods (Wheatley et al, and Pelleau et al). Indeed, we can see the same bi-phasic exponential pattern in Figure 1 and Figure 3a. The authors should consider fitting a bi-phasic exponential instead of an exponential model.

Response: In response to this comment and those above from reviewer 2 we have fitted 3 models – a linear (exponential), a biphasic exponential model and a spline-based model.

We agree that over the first 120 days after infection all 3 models are similar. We have therefore retained the linear model for the investigation of the covariates associated with antibody levels presented in new Table 2, as this yields the most interpretable coefficients.

However as suggested we now fit a biphasic exponential model for the purposes of determining longer term durations of antibody levels. This allows the flattening in the rates of antibody waning to be better represented. We have replaced new Figure 5 based on this model and updated the relevant parts of the text, and moved the figure based on the linear model to the supplement (Figure S11).

Minor point

I think figures of the style Figure S3 are much clearer and more informative than Figure 1. The authors could consider replacing Figure 1 with Figure S3.

Response: We have replaced Figure 1 with Figure S3.

References

Ainsworth, M. et al. Performance characteristics of five immunoassays for SARS-CoV-2: a head-to-head benchmark comparison. *The Lancet Infectious Diseases* 20, 1390–1400 (2020).

Lumley, S. F. et al. Antibody Status and Incidence of SARS-CoV-2 Infection in Health Care Workers. *New England Journal of Medicine* 384, 533–540 (2021).

Wheatley et al. Evolution of immune responses to SARS-CoV-2 in mild-moderate COVID-19. *Nature Comms.* 2021;

Pelleau et al. Kinetics of the SARS-CoV-2 antibody response and serological estimation of time since infection. *J Inf Dis.* 2021;

Reviewers' Comments:

Reviewer #2:

Remarks to the Author:

Thank you for this thorough revision. The responses to my specific comments were satisfactory, and I believe the content for which I can lend my expertise (chronic disease epidemiology) has improved and is now better represented. No study is perfect and the study limitations have been adequately addressed/discussed, including several sensitivity analyses made available only to reviewers. Best of luck in your future work.

Reviewer #3:

Remarks to the Author:

I think the authors have provided a considered and thoughtful response to the comments raised in my previous review.

My last small comment which the authors could address at their discretion (although I would recommend it) is to extend the timeframe of their predicted antibody levels in Sup Figure 10 from 120 days to 2100 days = the upper limit of time for which predictions are presented in Sup Figure 11. Note that of course it would not make sense to these predictions for the spline model.

Michael White

REVIEWERS' COMMENTS

Reviewer #2 (Remarks to the Author):

Thank you for this thorough revision. The responses to my specific comments were satisfactory, and I believe the content for which I can lend my expertise (chronic disease epidemiology) has improved and is now better represented. No study is perfect and the study limitations have been adequately addressed/discussed, including several sensitivity analyses made available only to reviewers. Best of luck in your future work.

Reviewer #3 (Remarks to the Author):

I think the authors have provided a considered and thoughtful response to the comments raised in my previous review.

My last small comment which the authors could address at their discretion (although I would recommend it) is to extend the timeframe of their predicted antibody levels in Sup Figure 10 from 120 days to 2100 days = the upper limit of time for which predictions are presented in Sup Figure 11. Note that of course it would not make sense to these predictions for the spline model.

Response: We have added two more panels to show the predicted antibody levels up to 2100 days for the linear exponential model and bi-phasic exponential model in Supplementary Fig. 10.